# MoCE: Mixture of Experts Representation for Robust Crystal Material Property Prediction

## Abstract

Accurately predicting the properties of crystal materials from their atomic structure is a fundamental challenge in materials science and computational chemistry. Graph-based models for crystal property prediction face a fundamental trade-off. They typically enforce a single geometric inductive bias, such as SE(3) invariance, which excels for periodic lattices but is less suited for materials where local chemistry and molecular conformation are dominant. To resolve this, we propose Mixture of Crystal Expert (MoCE) that dynamically integrates multiple, complementary geometric representations. Our model combines three specialized experts: an SE(3)-invariant module for global periodic structures, an SO(3)-invariant module for local atomic environments, and a dynamic graph module to learn latent topological interactions beyond fixed-bond assumptions. A gating network adaptively weighs each expert's contribution, tailoring the model's focus to the specific nature of a given crystal. This multi-representation approach achieves a more flexible and powerful framework, unifying the modeling of diverse crystal systems from rigid inorganic lattices to complex molecular crystals. Extensive validation on key benchmarks demonstrates state-of-the-art performance, confirming the effectiveness of our adaptive strategy.

## 1 Introduction

The computational design of novel materials hinges on our ability to accurately and efficiently predict properties from their underlying atomic structures (Ramprasad et al., 2017). In recent years, deep learning, particularly graph-based models (Yu et al., 2022; Gao & Ji, 2019; Fu et al., 2024), has emerged as a powerful paradigm for learning these structure-property relationships, offering a compelling alternative to the high computational cost of first-principles simulations like Density Functional Theory (DFT) (Xie & Grossman, 2018; Schütt et al., 2017). A key to their success lies in embedding specific geometric inductive biases, such as invariance or equivariance to Euclidean symmetries to learn effective representations of atomic environments (Batzner et al., 2022; Thomas et al., 2018). However, the vast chemical and structural diversity of materials, which spans from highly ordered inorganic lattices to flexible molecular crystals and disordered systems, suggests that any single, fixed geometric prior is insufficient.

This challenge creates a fundamental dichotomy in model design. Crystal material models have achieved great success by enforcing geometric invariance, successfully handling property prediction tasks (Choudhary & DeCost, 2021; Ito et al., 2025; Taniai et al., 2024). They typically construct a global representation of the unit cell and its infinite periodicity, prioritizing long-range interactions and the overall crystal lattice structure. Although effective for rigid inorganic crystals, this global focus becomes a limitation for a significant portion of materials, such as molecular or hybrid crystals, where properties are more heavily influenced by local atomic environments and intramolecular conformations. However, state-of-the-art models for molecular property prediction excel by meticulously capturing local chemistry, bonding, and conformational degrees of freedom, largely without considering long-range periodicity (Gasteiger et al., 2021; Shi et al., 2022; Gasteiger et al., 2020). An approach optimized for one regime fundamentally struggles to capture the essential physics of the other.

To resolve this tension, we propose a novel Mixture-of-Experts (MoE) framework that integrates multiple, complementary geometric representations within a single, unified model. Our approach,

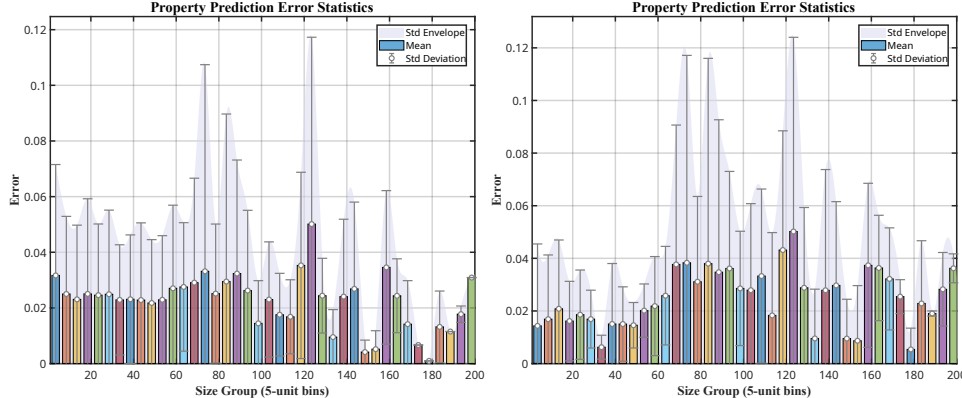

Figure 1: Statistical results of prediction errors in formation energy for different invariance models under identical architectures and settings on the MP dataset (SE(3)-invariant on the left, SO(3)-invariant on the right). The results indicate that models with different invariance properties exhibit distinct representational capabilities for crystal materials with varying numbers of atom per unit cell.

the Mixture of Crystal Experts (MoCE), dynamically combines the predictions of specialized expert networks: an SE(3)-invariant module to capture long-range periodic features, an SO(3)-invariant branch for local atomic environments, and a dynamic graph module for latent topological interactions. By adaptively weighting the contributions of these experts, our model can tailor its predictive strategy to the specific characteristics of a given material. We evaluated MoCE on widely adopted benchmark datasets for Crystal materials, and its superior performance across these diverse tasks demonstrates the effectiveness and generalization of our approach.

## 2 RELATED WORK

Effectively encoding the three-dimensional geometry, periodic nature, and underlying formation principles of Crystal materials is fundamental to accurately predicting their properties. Existing models can be broadly categorized into two main paradigms: invariant and equivariant architectures. Invariant models, a robust and widely adopted approach, operate on geometric quantities like interatomic distances and angles that are unchanged by rotations, ensuring inherent rotational invariance. In contrast, equivariant networks process features such as vectors and tensors that transform predictably with the system's geometry, enabling the direct modeling of directional properties like forces and achieving high accuracy, albeit often with increased computational complexity. Reviewing these efforts has helped us better understand how to leverage the advantages of both dynamic and static frameworks.

**Invariant-feature models.** A widely adopted approach builds message passing on inherently invariant geometric quantities so predictions remain unchanged under global rigid motions and translations. Representative crystal graph models CGCNN (Xie & Grossman, 2018) and MEGNet (Chen et al., 2019) encode interatomic distances and composition or chemistry to predict scalar properties. To better capture multi-body interactions, angle and direction-aware methods introduce triplet features and spherical-basis embeddings, as in DimeNet (Gasteiger et al., 2020), GemNet (Gasteiger et al., 2021), ALIGNN (Choudhary & DeCost, 2021) (via line-graph angles), and M3GNet (Chen & Ong, 2022) (higher-order interactions), which improve accuracy but can increase computational cost as neighbor counts and system sizes grow. More recently, PotNet (Lin et al., 2023) proposed physically motivated invariant edge features via periodic summations of predefined interatomic scalar potentials, offering a richer alternative to raw distances in crystals. While invariant feature models are robust and data-efficient, their representational capacity can be limited relative to architectures that explicitly model directional information or higher-order geometric responses (Pozdnyakov & Ceriotti, 2022).

**Equivariant-feature models.** Group-equivariant neural networks encode features that transform predictably under rotations and translations, enabling direct modeling of vector and tensor quantities while yielding invariant scalar outputs. Early frameworks such as Tensor Field Networks and Cormorant leveraged spherical harmonics and tensor representations to realize SE(3)/O(3)-equivariant

message passing for molecules and point clouds (Thomas et al., 2018). Subsequent models (Fuchs et al., 2020; Schütt et al., 2021) refined nonlinearity, efficiency, and scalability with learned vector features, attention mechanisms, and factorized filters. These methods can directly represent forces and directional interactions and deliver strong accuracy in atomistic modeling. However, strict equivariance often entails specialized nonlinearities and nontrivial computational overhead, and accommodating periodic boundary conditions and unit-cell variations in crystals adds design complexity beyond finite molecular systems. Efficient designs are increasingly using local frames and linear computations in neighbor size to balance accuracy and cost in large structures (Musaelian et al., 2023).

**Dynamic topology.** Moving beyond fixed graph construction, dynamic topology adaptively captures specific local interactions. Although CrystalFramer (Ito et al., 2025) employs attention to build task-aware local environments, it remains constrained by an underlying static graph, a limitation absent in graph learning methods that explicitly infer connectivity. The Differentiable Graph Module (DGM) (Kazi et al., 2023) learns edge probabilities end-to-end, and (Sun et al., 2024) introduces learnable pseudo-nodes for efficient dynamic messaging. (Zohrabi et al., 2024) extends neighborhoods via centrality or similarity. These methods collectively achieve more expressive graph representations, providing ideas for designing new representation methods that are beneficial for disordered or heterogeneous materials.

## 3 PRELIMINARY

The crystal structure is formally represented as $\mathcal{G} = (\mathbf{A}, \mathbf{P}, \mathbf{L})$, where $\mathbf{A} = [a_1, ..., a_N]^\top$ denotes atomic species, $\mathbf{P} = [\mathbf{p}_1, ..., \mathbf{p}_N]^\top$ their Cartesian coordinates, and $\mathbf{L} = [\mathbf{l}_1, \mathbf{l}_2, \mathbf{l}_3]^\top$ the lattice vectors defining the unit cell (details provided in Appendix A). This representation forms the basis of graph-based deep learning for material property prediction, making it fundamentally different from traditional methods.

**Definition 1** (Density Functional Theory (DFT) (Kohn et al., 1996)) In DFT, the total energy of a crystal is derived from the electron density $n(\mathbf{r})$ within a periodic potential. This is formulated through the Kohn–Sham equations:

$$\left( -\frac{\hbar^2}{2m} \nabla^2 + V_{\text{eff}}[n](\mathbf{r}) \right) \psi_i(\mathbf{r}) = \epsilon_i \psi_i(\mathbf{r}), \tag{1}$$

where the effective potential $V_{\text{eff}}$ depends on both the electron density and the positions of the ions. $\psi_i(\cdot)$ is the single-particle wavefunction.

**Definition 2** (Geometrically Complete Crystal Graph (Widdowson & Kurlin, 2022)) A crystal graph $G$ is geometrically complete if the equality $G_1 = G_2$ implies that the corresponding infinite crystals $M_1$ and $M_2$ are isometric, denoted $M_1 \cong M_2$. That is, identical graphs must represent identical atomic structures up to isometry.

**Key Challenges.** Predicting crystal properties via graph-based machine learning introduces challenges that arise from the macroscopic abstraction inherent in graph representations, in contrast to the first-principles rigor of methods such as DFT. While DFT is grounded in fundamental quantum mechanical equations, graph-based approaches require the compression of an infinite, periodic system into a finite, learnable structure. This process inevitably involves information loss and adds considerable complexity. Consequently, it necessitates not only the imposition of strict constraints on graph representation but also a flexible learning framework capable of capturing underlying physical principles.

Moreover, unlike finite molecules, crystals are periodic structures defined by a unit cell and lattice vectors $[\mathbf{l}_1, \mathbf{l}_2, \mathbf{l}_3]$. Machine learning models must generalize across diverse space groups and chemical compositions, which requires robust incorporation of crystal symmetries that are not fully captured in existing graph architectures. Furthermore, many material properties of interest—such as elasticity and thermal conductivity—are tensorial in nature and depend on crystal orientation. Their prediction therefore demands equivariant representations that go beyond the capabilities of simple graph encoders.

Previous studies (Yan et al., 2024; Liu et al., 2022) have begun to address these issues by incorporating the unique passive symmetries arising from the multiplicity of unit cell choices and periodic

Figure 2: The overview of the proposed framework constructed with a SE(3)-invariant (distance/angle) expert, a SO(3)-invariant (angle) expert, and a dynamic graph expert. Each representation is processed by a specialized network. A Mixture of Experts module then intelligently weighs and fuses these distinct geometric and topological features to produce a final, robust prediction.

boundary conditions. These include enforcing invariance or equivariance at the unit cell level, which are essential for consistent and accurate representation of crystal structures.

## 4 MIXTURE OF EXPERTS CRYSTAL STRUCTURE REPRESENTATION

As illustrated in Fig. 2, we propose a mixture-of-experts architecture that incorporates multiple geometric representations. It consists of three expert branches: an SO(3)-invariant module for global properties, an SE(3)-invariant module for angular details, and a differentiable GNN for adaptive graph reasoning. A gating network dynamically combines these experts, enabling flexible and input-dependent representation learning for robust performance across diverse material systems.

### 4.1 DIVERSITY OF CRYSTAL STRUCTURE REPRESENTATION

The vast chemical and structural diversity of crystal materials poses a fundamental challenge to representation learning. A prevailing assumption is that a single, monolithic architecture with a fixed inductive bias can serve as an optimal encoder $f : \mathcal{X} \to \mathbb{R}^d$ for the entire material space $\mathcal{X}$. We argue that this is insufficient due to the need to reconcile multiple, often competing, symmetries and scales inherent to materials science.

**Definition 3** (Global crystal Symmetry). A perfect crystal structure $M$ is defined by a motif and a lattice, belonging to a space group $S$. The space group is the set of all symmetry operations $g = \mathbf{R}|\mathbf{t}$ that leave the infinite crystal invariance, where $\mathbf{R}$ is a point group operation (rotation, reflection, inversion) and $\mathbf{t}$ is a translation. An ideal model $f(\cdot)$ must satisfy invariance under all $g \in S$: $f(M) = f(g \circ M)$.

**Definition 4** (Local Euclidean Invariance (Thomas et al., 2018; Brandstetter et al., 2021)). For predicting scalar properties (e.g., formation energy) from local atomic environments, a representation must be invariant to global rotations and translations, i.e., **SE(3)-invariant**: $f(\mathbf{X}) = f(\mathbf{RX} + \mathbf{t})$ for any $\mathbf{R} \in SO(3)$ and translation $\mathbf{t}$.

**Definition 5** (Molecular Unit Invariance). In molecular crystals, properties may depend on the internal conformation of a molecule, which is decoupled from its global orientation in the lattice. This requires invariance only to global rotations (**SO(3)-invariance**) of the entire system: $f(\mathbf{X}) = f(\mathbf{RX})$ for any $\mathbf{R} \in SO(3)$.

**Definition 6** (Topological Sensitivity). For disordered or flexible systems, long-range covalent connectivity defines the material's identity. A static graph $\mathcal{G}_{\text{static}} = (\mathcal{V}, \mathcal{E}_{\text{cutoff}})$, constructed from a

single snapshot using a cutoff radius $r_c$, may fail to capture the true covalent topology $\mathcal{G}_{\text{covalent}}$: $\mathcal{G}_{\text{static}} \not\equiv \mathcal{G}_{\text{covalent}}$. A representation sensitive to this dynamic graph structure is required.

Existing models excel in narrow regimes. SE(3)-invariant networks (Batzner et al., 2022; Taniai et al., 2024), which typically operate in local neighborhoods $N(i)$ defined by a cutoff, are well-suited for contexts requiring local Euclidean invariance, but are often blind to scenarios involving topological sensitivity. Purely invariant models designed for contexts governed by both local Euclidean and molecular unit invariances are fundamentally limited in representing systems defined by continuous deformations or dynamics, as they discard all but the invariant features.

## 4.2 SE(3)-INVARIANT CRYSTAL STRUCTURE REPRESENTATION

In geometric graph neural networks for crystal materials, invariance is essential, as crystal properties remain unchanged under rigid rotations and translations. The invariant branch encodes local atomic environments using invariant geometric features within coordination spheres (Liu et al., 2022), ensuring consistent and transferable representations for reliable periodic properties of crystals (Yan et al., 2024).

**Definition 7** (Special Euclidean Group SE(3)). An SE(3) transformation $g \in \text{SE}(3)$ acts on the atomic coordinates as $g \circ \mathbf{p}_i = \mathbf{R}\mathbf{p}_i + \mathbf{t}$, where $\mathbf{R} \in \mathbb{R}^{3 \times 3}$ is a rotation matrix ($\mathbf{R}^{\top}\mathbf{R} = \mathbf{I}, \det(\mathbf{R}) = 1$) and $\mathbf{t} \in \mathbb{R}^3$ is a translation vector.

The basic architecture of graph neural networks has evolved from the previous graph convolution (Xie & Grossman, 2018; Schütt et al., 2018) to graph transformers (Yan et al., 2024; Ito et al., 2025). To construct an SE(3)-invariance $f(\mathbf{X}, \mathbf{R}\mathbf{P} + \mathbf{t}, \mathbf{R}\mathbf{L}) = f(\mathbf{X}, \mathbf{P}, \mathbf{L}) \quad \forall g = (\mathbf{R}, \mathbf{t}) \in \text{SE}(3)$, the message-passing update for an atom's state in an SE(3)-invariant transformer is therefore defined as:

$$\mathbf{x}_i^{(t+1)} = \phi\left(\mathbf{x}_i^{(t)}, \bigoplus_{j \in \mathcal{N}(i)} \psi\left(\mathbf{x}_i^{(t)}, \mathbf{x}_j^{(t)}, \|\mathbf{e}_{ij}\|_2, \ldots\right)\right), \tag{2}$$

where $\phi$ and $\psi$ are learned functions, $\bigoplus$ is a permutation-invariant aggregation operator (e.g., sum or mean), and $(\ldots)$ indicates that other invariant edge attributes (e.g., distance, triplet angles) may also be incorporated. By construction, this update rule satisfies **Definition 7**, as it depends only on invariant scalars and the already-invariant atom features $\mathbf{x}$.

Specifically, each atom $i$ is associated with an initial feature vector $\mathbf{x}_i^{(0)} \in \mathbb{R}^d$, typically obtained by embedding a layer that maps atomic numbers to a continuous space $\mathbf{x}_i^{(0)} \leftarrow \text{Embedding}(a_i)$. The primary invariants are the interatomic distance $\|\mathbf{e}_{ij}\|_2$ and the angles $\{\theta_{ij}^{(1)}, \theta_{ij}^{(2)}, \theta_{ij}^{(3)}\}$ between edges $\mathbf{e}_{ij}$ and lattice representation $\{\mathbf{e}_{ii1}, \mathbf{e}_{ii2}, \mathbf{e}_{ii3}\}$. For a pair of atoms $(i, j)$, the Euclidean distance $\|\mathbf{e}_{ij}\|_2 = \|\mathbf{p}_i - \mathbf{p}_j\|$ is invariant to both rotation and translation. Under periodic boundary conditions, this distance is computed using infinitely connected distance-decay attention (Taniai et al., 2024). The attention mechanism models interatomic interactions by updating the state of atom $i$, $\mathbf{x}_i'$, as a weighted sum over all atoms $j$ and their periodic images $\boldsymbol{n}$:

$$\boldsymbol{x}_i' = \frac{1}{Z_i} \sum_{j=1}^{N} \sum_{\boldsymbol{n} \in \mathbb{Z}^3} \exp\left(\frac{\boldsymbol{q}_i^T \boldsymbol{k}_j}{\sqrt{d_K}} - \frac{\|\boldsymbol{p}_{j(\boldsymbol{n})} - \boldsymbol{p}_i\|^2}{2\sigma_i^2}\right)(\boldsymbol{v}_j + \boldsymbol{\psi}_{ij(\boldsymbol{n})}), \tag{3}$$

where the attention weight simultaneously captures feature similarity through the scaled dot-product of query $\boldsymbol{q}_i$ and key $\boldsymbol{k}_j$, and spatial locality through a Gaussian distance-decay term. This decay term, governed by the adaptively learned width $\sigma_i$, ensures that the influence of an atom diminishes with its distance $\|\boldsymbol{p}_{j(\boldsymbol{n})} - \boldsymbol{p}_i\|$. When the value of $\sigma_i$ is not too large, the infinite series $\sum_{\boldsymbol{n}}$ will rapidly converge to the upper bound of the error (Taniai et al., 2024). Furthermore, the value term is enhanced with a geometric position embedding $\boldsymbol{\psi}_{ij(\boldsymbol{n})}$ that explicitly encodes this interatomic distance and angle information. The summations over $j$ and $\boldsymbol{n}$ represent all atoms in the unit cell and their periodic images, respectively, while $Z_i = \sum_{j=1}^{N} \sum_{\boldsymbol{n}} \exp\left(\boldsymbol{q}_i^T \boldsymbol{k}_j / \sqrt{d_K} - \|\boldsymbol{p}_{j(\boldsymbol{n})} - \boldsymbol{p}_i\|^2 / 2\sigma_i^2\right)$ is the normalization factor.

To obtain the angle-invariant embedding $\boldsymbol{\psi}_{ij(\boldsymbol{n})}$, we encode the relative position vector $\vec{r}_{ij}(\boldsymbol{n}) = \vec{p}_j(\boldsymbol{n}) - \vec{p}_i(\boldsymbol{n})$. This is achieved by combining features based on both distance and orientation. For the orientation, the direction vector $\vec{r}_{ij}(\boldsymbol{n})$ is projected onto the lattice representation $\{\mathbf{e}_{ii1}, \mathbf{e}_{ii2}, \mathbf{e}_{ii3}\}$. This results in three scalar components, $\theta_{ij}^{(k)}(\boldsymbol{n})$, each representing the cosine of the angle between $\vec{r}_{ij}(\boldsymbol{n})$ and the corresponding axis $e_{iik}$.

These scalar angle components, along with the distance scalar $r_{ij}(\boldsymbol{n}) = ||\vec{r}_{ij}(\boldsymbol{n})||$, are then encoded into high-dimensional vectors using Gaussian Basis Functions (GBFs). The GBF expansion maps a scalar input $x$ to a $D$-dimensional vector $\mathbf{b}(x)$, where the $k$-th component is computed as a Gaussian:

$$b_k(x) = \exp\left(-\frac{(x - \mu_k)^2}{2\sigma_k^2}\right),\tag{4}$$

Here, the means $\{\mu_k\}_{k=1}^n$ are uniformly distributed within a range $[\mu_{min}, \mu_{max}]$, and the widths $\sigma_k$ are proportional to this interval.

The final geometric relative position encoding $\psi_{ij}(\boldsymbol{n})$ is constructed by a learnable linear projection of the distance-based and angle-based GBF vectors:

$$\psi_{ij}(\boldsymbol{n}) = W_0 \mathbf{b}_{\text{dist}}(r_{ij}(\boldsymbol{n})) + \sum_{k=1}^{3} W_k \mathbf{b}_{\text{angle}}(\theta_{ij}^{(k)}(\boldsymbol{n})),\tag{5}$$

where $\{W_0, W_1, W_2, W_3\}$ are trainable weight matrices. For the angle embeddings, we use a cosine range of $[-1.0, 1.0]$ and set the GBF hyper-parameters $\{\mu_{min}, \mu_{max}, s, D\}$ to $\{-1.0, 1.0, 4.0, 64\}$ following (Ito et al., 2025).

## 4.3 SO(3)-INVARIANT CRYSTAL STRUCTURE REPRESENTATION

The SO(3)-invariant expert is engineered to capture geometric features distinct from those learned by the SE(3)-invariant branch. While the SE(3) expert targets the translational and rotational symmetries inherent to extended periodic lattices, the SO(3) expert focuses exclusively on rotational invariance. This deliberate change in symmetry constraint makes it specialized for modeling the intrinsic geometry and conformational degrees of freedom of the constituent molecular units within a crystal (Musil et al., 2021).

This architecture is particularly adept at representing molecular crystals, where macroscopic properties are often dominated by the intramolecular configuration of the molecules rather than their long-range crystallographic packing (Reilly et al., 2016). For example, properties such as vibrational frequencies and optical activity are strongly tied to the molecule's internal bond lengths, angles, and torsions. By operating on a representation that is invariant to molecular orientation but sensitive to conformation, this branch effectively decouples the internal geometry of the molecule from its position and orientation within the lattice. This provides a specialized feature space that complements the periodic pattern recognition of the SE(3) branch, enabling our model to form a more holistic, multiscale understanding of the material.

In terms of specific implementation, the differences in the SO(3) invariance branch mainly lie in the processing of edge information. The SO(3) invariance requires the network to ignore rotation information. Therefore, compared to Eq. 3, the calculation of attention removes the $\boldsymbol{\psi}_{ij(\boldsymbol{n})}$ (i.e., the angle encoding) term, defined as:

$$\boldsymbol{x}_i' = \frac{1}{Z_i} \sum_{j=1}^{N} \sum_{\boldsymbol{n} \in \mathbb{Z}^3} \exp\left(\frac{\boldsymbol{q}_i^T \boldsymbol{k}_j}{\sqrt{d_K}} - \frac{\Theta(\|\boldsymbol{p}_{j(\boldsymbol{n})} - \boldsymbol{p}_i\|^2)}{2\sigma_i^2}\right) \boldsymbol{v}_j,\tag{6}$$

where $\Theta(\cdot)$ denotes learnable linear layers and active functions, providing non-linearity that enables the distance not to be confined to Euclidean space.

## 4.4 DYNAMIC GRAPH CRYSTAL STRUCTURE REPRESENTATION

While geometric representations based on Euclidean distances and angles are powerful, they are predicated on a predefined and static graph construction, typically based on a radial cutoff. This

heuristic may not optimally represent the complex, nonlocal, and often subtle bonding interactions present in diverse crystal structures (Gasteiger et al., 2021). To address this limitation and capture latent connectivity patterns beyond fixed spatial heuristics, we introduce a Differentiable Dynamic Graph Network (Kazi et al., 2023). This module learns to generate a task-specific graph structure directly from atomic features, allowing for a more flexible and expressive representation of atomic interactions.

Due to the relationship between two atoms, $i$ and $j$ can be inferred from their features, the Dynamic Graph Generation (DGM) module projects the node features into a latent space and computes a cross attention.

$$\mathbf{A}_{\text{dynamic}} = \text{Softmax}(\frac{\boldsymbol{q}_i^T \boldsymbol{k}_j}{\sqrt{d_K}}), \tag{7}$$

The dynamically generated graph captures latent relationships, while the initial static fully connection graph (based on a unit cell) encodes fundamental spatial proximity. To take advantage of both, we fuse them. Let $\mathbf{A}_{\text{static}}$ be the adjacency matrix of the initial graph, the fused graph adjacency matrix, $\mathbf{A}_{\text{fused}}$, is obtained through a weighted summation:

$$\mathbf{A}_{\text{fused}} = \alpha \mathbf{A}_{\text{static}} + (1 - \alpha)\sigma(\mathbf{A}_{\text{dynamic}}), \tag{8}$$

where $\sigma(\cdot)$ is the sigmoid function applied element-wise to normalize the dynamic adjacency matrix, and $\alpha \in [0, 1]$ is a learnable gating parameter that adaptively balances the influence of the static and learned graph structures.

Operating on a dense graph is computationally prohibitive, and most of the learned connections in $\mathbf{A}_{\text{dynamic}}$ may be spurious. Therefore, we enforce sparsity to retain only the most salient connections. A new sparse adjacency matrix, $\mathbf{A}_{\text{new}}$, is generated by applying a top-k selection operator or a threshold-based sparsification on $\mathbf{A}_{\text{fused}}$. For instance, by using top-k, we have

$$A_{\text{new}}^{(ij)} = \begin{cases} A_{\text{fused}}^{(ij)}, & \text{if } A_{\text{fused}}^{(ij)} \in \text{Top-K}(\mathbf{A}_{\text{fused},i}), \\ 0, & \text{otherwise,} \end{cases} \tag{9}$$

where $\text{Top-K}(\mathbf{A}_{\text{fused},i})$ denotes the set of the $k$ largest values in the $i$-th row of $\mathbf{A}_{\text{fused}}$. This entire process—generation, fusion, and sparsification—is differentiable end-to-end, enabling the graph structure itself to be optimized via gradient descent for the downstream property prediction task (Kazi et al., 2023). The dynamic graph generation module is conducted at the beginning of each layer in GNN.

## 4.5 MIXTURE-OF-EXPERTS FOR FUSED REPRESENTATION

To synthesize the complementary representations learned by the SE(3)-invariant, SO(3)-invariant, and DGMGNN branches, we employ a Mixture-of-Experts (MoE) framework (Shazeer et al., 2017). This module adaptively weights the contributions of each expert based on the input data, allowing the model to emphasize the most relevant features for a given crystal structure.

Let $\mathbf{x}_{\text{SE(3)}}^{(n)}$, $\mathbf{x}_{\text{SO(3)}}^{(n)}$, and $\mathbf{x}_{\text{DG}}^{(n)}$ be the final graph-level representations obtained by pooling the node embeddings from each of the three expert branches. A lightweight gating network, parameterized as a simple multi-layer perceptron (MLP), computes a distribution over the experts from a shared input representation, such as the concatenated initial features $[\mathbf{x}_{\text{SE(3)}}^{(n)}; \mathbf{x}_{\text{SO(3)}}^{(n)}; \mathbf{x}_{\text{DG}}^{(n)}]$.

$$g(\mathbf{x}) = \text{softmax}(\text{MLP}([\mathbf{x}_{\text{SE(3)}}^{(n)}; \mathbf{x}_{\text{SO(3)}}^{(n)}; \mathbf{x}_{\text{DG}}^{(n)}])). \tag{10}$$

The output of the gating network is a vector of weights, $\mathbf{w} = [w_{\text{SE(3)}}, w_{\text{SO(3)}}, w_{\text{DG}}]$, where $\sum_i w_i = 1$. The final, fused representation for the crystal, $\mathbf{x}_{\text{final}}$, is then computed as the weighted sum of the expert outputs:

$$\mathbf{x}_{\text{final}} = w_{\text{SE(3)}}\mathbf{x}_{\text{SE(3)}}^{(n)} + w_{\text{SO(3)}}\mathbf{x}_{\text{SO(3)}}^{(n)} + w_{\text{DG}}\mathbf{x}_{\text{DG}}^{(n)}. \tag{11}$$

This aggregated vector $\mathbf{x}_{\text{final}}$ is then passed to a final prediction head, typically another MLP, to regress the target material property. This MoE architecture enables the model to learn a flexible, data-driven policy for combining diverse structural and relational information, enhancing its expressive power and generalization capability across varied material classes.

Table 1: The results of the property predictions on **MP** in terms of MAE. The best results is indicated in **bold**, and the second-best results is marked with an underline.

| Method | Form. energy (eV/atom) 60000 / 5000 / 4239 | Bandgap (eV) 60000 / 5000 / 4239 | Bulk modulus (log(GPa)) 4664 / 393 / 393 | Shear modulus (log(GPa)) 4664 / 392 / 393 |
|---|---|---|---|---|
| CGCNN (Xie & Grossman, 2018) | 0.031 | 0.292 | 0.047 | 0.077 |
| SchNet (Schütt et al., 2018) | 0.033 | 0.345 | 0.066 | 0.099 |
| MEGNet (Chen et al., 2019) | 0.030 | 0.307 | 0.060 | 0.099 |
| GATGNN (Louis et al., 2020) | 0.033 | 0.280 | 0.045 | 0.075 |
| M3GNet (Chen & Ong, 2022) | 0.024 | 0.247 | 0.050 | 0.087 |
| ALIGNN (Choudhary & DeCost, 2021) | 0.022 | 0.218 | 0.051 | 0.078 |
| Matformer (Yan et al., 2022) | 0.021 | 0.211 | 0.043 | 0.073 |
| PotNet (Lin et al., 2023) | 0.0188 | 0.204 | 0.040 | 0.065 |
| eComFormer (Yan et al., 2024) | 0.0182 | 0.202 | 0.0417 | 0.0729 |
| iComFormer (Yan et al., 2024) | 0.0183 | 0.193 | 0.038 | 0.0637 |
| Crystalformer (Taniai et al., 2024) | 0.0186 | 0.198 | 0.0377 | 0.0689 |
| CrystalFramer (PCA) (Ito et al., 2025) | 0.0197 | 0.214 | 0.0423 | 0.0715 |
| CrystalFramer (MAX) (Ito et al., 2025) | 0.0172 | 0.185 | 0.0338 | 0.0677 |
| MoCE (Proposed) | **0.0161** | **0.162** | **0.0325** | **0.0631** |

Table 2: The results of the property predictions on **JARVIS** in terms of MAE.

| Method | Form. energy (eV/atom) 44578 / 5572 / 5572 | Total energy (eV/atom) 44578 / 5572 / 5572 | Bandgap (OPT) (eV) 44578 / 5572 / 5572 | Bandgap (MBJ) (eV) 14537 / 1817 / 1817 | E hull (eV) 44296 / 5537 / 5537 |
|---|---|---|---|---|---|
| CGCNN (Xie & Grossman, 2018) | 0.063 | 0.078 | 0.20 | 0.41 | 0.17 |
| SchNet (Schütt et al., 2018) | 0.045 | 0.047 | 0.19 | 0.43 | 0.14 |
| MEGNet (Chen et al., 2019) | 0.047 | 0.058 | 0.145 | 0.34 | 0.084 |
| GATGNN (Louis et al., 2020) | 0.047 | 0.056 | 0.17 | 0.51 | 0.12 |
| M3GNet (Chen & Ong, 2022) | 0.039 | 0.041 | 0.145 | 0.362 | 0.095 |
| ALIGNN (Choudhary & DeCost, 2021) | 0.0331 | 0.037 | 0.142 | 0.31 | 0.076 |
| Matformer (Yan et al., 2022) | 0.0325 | 0.035 | 0.137 | 0.30 | 0.064 |
| PotNet (Lin et al., 2023) | 0.0294 | 0.032 | 0.127 | 0.27 | 0.055 |
| eComFormer (Yan et al., 2024) | 0.0284 | 0.032 | 0.124 | 0.28 | 0.044 |
| iComFormer (Yan et al., 2024) | 0.0272 | 0.0288 | 0.122 | 0.26 | 0.047 |
| Crystalformer (Taniai et al., 2024) | 0.0306 | 0.032 | 0.128 | 0.274 | 0.0463 |
| CrystalFramer (PCA) (Ito et al., 2025) | 0.0287 | 0.0305 | 0.126 | 0.279 | **0.0444** |
| CrystalFramer (MAX) (Ito et al., 2025) | 0.0263 | 0.0279 | 0.117 | 0.242 | 0.0471 |
| MoCE (Proposed) | **0.0243** | **0.0257** | **0.112** | **0.239** | 0.0445 |

## 5 EXPERIMENTS

We evaluate our model on three standard materials science benchmarks: the Joint Automated Repository for Various Integrated Simulations (JARVIS), the Materials Project (MP), and the Open Quantum Materials Database (OQMD), predicting key properties such as formation energy, bandgap, and bulk modulus. Data splits follow established protocols from comparison work( (Ito et al., 2025; Yan et al., 2024; Taniai et al., 2024)). The model was trained for 1000 epochs using MAE loss and the AdamW optimizer, with a batch size of 256 and an initial learning rate of 5e-4 decayed inversely. We applied Stochastic Weight Averaging (SWA) over the last 50 epochs and selected the best validation checkpoint for testing. All experiments were run on NVIDIA A6000 GPUs. We compare our proposed method with previous learning-based state-of-the-art methods (Xie & Grossman, 2018; Schütt et al., 2018; Chen et al., 2019; Louis et al., 2020; Chen & Ong, 2022; Choudhary & DeCost, 2021; Yan et al., 2022; Lin et al., 2023; Yan et al., 2024; Taniai et al., 2024; Ito et al., 2025). Further details of the implementation are provided in Appendix B.

### 5.1 RESULTS

**Property Prediction Performance Comparison.** We benchmarked our proposed Mixture of Crystal Experts (MoCE) model against a comprehensive suite of state-of-the-art models on the MP, JARVIS, and OQMD datasets. The comparative results, measured in Mean Absolute Error (MAE), demonstrate the consistent superiority of our method. Our MoCE establishes new state-of-the-art performance on the vast majority of prediction tasks across all three benchmarks. As shown in Table 1, our method attains state-of-the-art performance in the prediction of these properties. In particular, the most significant enhancements are observed for properties characterized by a relatively larger sample size. This robust performance extends to the challenging JARVIS dataset (Table 2), confirming the effectiveness and scalability of our model. As shown in Table 3, our validation results on the large-scale OQMD dataset demonstrate the predictive stability and data scalability of our method under complex and diverse data conditions. These results strongly validate our central hypothesis. By integrating multiple complementary geometric representations, including SE(3)-invariant, SO(3)-invariant, and dynamic graph-based experts, the MoE framework captures a richer

Table 3: The results of the property predictions on **OQMD** in terms of MAE.

| Method | # Blocks | Form. energy (eV/atom) 654108 / 81763 / 81763 | Bandgap (eV) 653388 / 81673 / 81673 | E hull (eV/atom) 654108 / 81763 / 81763 |
|---|---|---|---|---|
| Crystalformer | 4 | 0.02115 | 0.06028 | 0.06759 |
| CrystalFramer | 4 | 0.01871 | 0.05805 | 0.06607 |
| CrystalFramer-lightweight | 4 | 0.01813 | 0.05773 | 0.06672 |
| MoCE (Proposed) | 4 | **0.01692** | **0.05638** | **0.06481** |

structure-property relationship. It successfully adapts to diverse crystal systems by dynamically weighing expert contributions, overcoming the limitations of models reliant on a single geometric inductive bias and yielding more accurate, generalizable predictions across vast chemical spaces.

**Efficiency Analysis.** While our MoCE model achieves competitive predictive performance, its computational efficiency presents certain limitations. As shown in Table 4, MoCE exhibits higher latency (116 ms) and a larger parameter count (5.7M) compared to several recent methods. This overhead stems primarily from the multi-expert architecture and the dynamic graph operations, which involve sequential computation steps, such as iterative edge updates and structure-dependent graph construction that limit parallelization and increase memory usage. Although these operations enhance representational flexibility, they introduce recurrent costs that scale with graph complexity. Future

Table 4: Efficiency analysis.

| Method | Latency | #Params | #Params/Block |
|---|---|---|---|
| PotNet | 343ms | 1.8M | 527K |
| Matformer | 24ms | 2.9M | 544K |
| iComFormer | 59ms | 5M | 855K |
| Crystalformer | 11ms | 853K | 806K |
| CrystalFramer | 20ms | 952K | 231K |
| MoCE | 116ms | 5.7M | 901K |

work will explore optimizations such as cached graph descriptors, approximate attention mechanisms, and partially precomputed latent graphs to reduce sequential dependence and improve computational throughput.

**Ablation Study.** To dissect the contribution of each component within our architecture, we conducted a thorough ablation studies on the formation energy and bandgap prediction tasks on the JARVIS dataset. As shown in 5, removing any single expert leads to a noticeable degradation in performance for both formation energy and bandgap prediction. The most significant performance drop occurs upon removal of the SE(3)-invariant expert, highlighting its critical role in capturing local atomic environments and complete periodic representation of crystals. However, ablating the SO(3)-invariant and dynamic graph modules also impairs accuracy, confirming their value in representing molecular conformations and latent bonding, respectively. Furthermore,

Table 5: Ablation Study on JAVIS dataset.

| Method | Form. energy (eV/atom) | Bandgap (eV) |
|---|---|---|
| MoCE | **0.0243** | **0.112** |
| w/o SE(3)-invariant | 0.0312 | 0.134 |
| w/o SO(3)-invariant | 0.0279 | 0.123 |
| w/o Dynamic graph | 0.0266 | 0.115 |
| w/o MoE | 0.0257 | 0.121 |

the variant without the MoE gating mechanism, which is replaced by a simpler aggregation, performs significantly worse than the full MoCE model. Specifically, as shown in Appendix C Fig. 3, the proposed framework not only enhances performance, but also improves the generalization for different types of materials. It validates the effectiveness of our adaptive gating network, which learns to intelligently weigh and combine the complementary geometric representations based on the input crystal structure, thereby achieving a more powerful and flexible representation.

## 6 CONCLUSION

In this work, we introduced a mixture-of-experts architecture that integrates complementary geometric representations (i.e., SE(3)-invariance, SO(3)-invariance, and latent dynamic graphs) to address the diverse symmetry and structural nature of Crystal materials. By adaptively combining these experts, our model achieves more accurate and generalizable property predictions across a wide range of crystal systems. While the approach incurs additional computational cost, the gains in expressive power and physical alignment justify the trade-off. Future work will focus on improving computational efficiency without sacrificing representational flexibility.

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

## A    REPRESENTATION OF CRYSTAL STRUCTURES

A crystal material is defined by its periodic atomic structure, which can be completely specified by a unit cell and the rules for its repetition in space. Following the established formalism (Yan et al., 2022), we represent a crystal structure as a triplet $\mathcal{G} = (\mathbf{A}, \mathbf{P}, \mathbf{L})$. The chemical elements (Atomic Species $\mathbf{A}$) of the $N$ atoms within the unit cell are given by a vector:

$$\mathbf{A} = [a_1, a_2, \ldots, a_N]^\top, \quad a_i \in \mathbb{N}. \tag{12}$$

Each $a_i$ corresponds to an atomic number. The Cartesian coordinates (Atomic Positions $\mathbf{P}$) of the atoms within the unit cell are stored in a matrix:

$$\mathbf{P} = [\mathbf{p}_1, \mathbf{p}_2, \ldots, \mathbf{p}_N]^\top, \quad \mathbf{p}_i \in \mathbb{R}^3. \tag{13}$$

The shape and size of the unit cell are defined by three lattice vectors (lattice vectors $\mathbf{L}$), which form a matrix:

$$\mathbf{L} = [\mathbf{l}_1, \mathbf{l}_2, \mathbf{l}_3]^\top \in \mathbb{R}^{3 \times 3}. \tag{14}$$

The infinite periodic crystal is generated by translating the unit cell by all integer linear combinations of the lattice vectors, $\mathbf{n} \in \mathbb{Z}^3$. The species and coordinates of any atom in this crystal are given by:

$$a_{i(\mathbf{n})} = a_i, \tag{15}$$

$$\mathbf{p}_{i(\mathbf{n})} = \mathbf{p}_i + \mathbf{L}\mathbf{n}, \tag{16}$$

where index $i(\mathbf{n})$ denotes the copy of the $i$-th atom generated by the translation $\mathbf{L}\mathbf{n}$ (Taniai et al., 2024). This representation $(\mathbf{A}, \mathbf{P}, \mathbf{L})$ forms the foundational input for crystal material prediction in graph-based methods.

## B  EXPERIMENT DETAILS

This section details the empirical evaluation of our proposed model. We outline the datasets used for benchmarking, describe the experimental setup and training parameters, and define the evaluation metrics.

### B.1  DATASETS

Our empirical validation is conducted on three widely recognized benchmark datasets for materials science: the Joint Automated Repository for Various Integrated Simulations (JARVIS), the Materials Project (MP), and the Open Quantum Materials Database (OQMD). Each dataset comprises a vast collection of crystal structures annotated with various properties derived from high-throughput Density Functional Theory (DFT) simulations.

**JARVIS** dataset, specifically the JARVIS-DFT 3D 2021 collection (Choudhary et al., 2020), provides a curated set of 55,723 materials. For this collection, properties have been simulated using multiple DFT calculation methods, including the OptB88vdW version (i.e., OPT) and the TBmBJ version (i.e., MBJ). Our regression targets from this dataset include key physical quantities such as formation energy per atom, electronic bandgap, and energy above the convex hull.

**Materials Project (MP)** database is another foundational resource in materials informatics, originally detailed by Jain et al. (Jain et al., 2013). We utilize a common snapshot containing 69,239 materials, as organized by Chen et al. (Chen et al., 2019). The properties of interest from MP include the formation energy and bandgap calculated with the PBE functional, as well as mechanical properties like bulk and shear moduli.

**Open Quantum Materials Database (OQMD)**, introduced by Kirklin et al. (Kirklin et al., 2015), is a large-scale database designed for thermodynamic stability analysis. Our work leverages a snapshot that encompasses 817,636 crystal structures. From this extensive dataset, we focus on predicting the formation energy, bandgap, and the energy above the convex hull.

For our experiments, we focus on predicting these key material properties chosen to align with prior work. To ensure a fair and rigorous comparison against existing methods, we adopt the standardized data partitioning for training, validation, and testing as established in previous benchmark studies (Ito et al., 2025; Yan et al., 2024; Taniai et al., 2024).

### B.2  IMPLEMENTATION DETAILS

The model parameters were optimized from a random initialization by minimizing the mean absolute error (MAE) loss function between the predicted and ground-truth values with the AdamW optimizer (Loshchilov & Hutter, 2017). The $(\beta 1, \beta 2)$ and weight decay are setting to $(0.9, 0.98)$ and 10e-5 respectively. The model was trained for a total of 1000 epochs with a batchsize of 256. The learning rate was dynamically adjusted throughout the training process. We employed an inverse square root decay schedule (Huang et al., 2020), with an initial learning rate set to 5e-4. To improve the model's generalization capabilities, we applied Stochastic Weight Averaging (SWA) (Izmailov et al., 2018) on the model weights from the final 50 epochs of training. The model checkpoint that yielded the best performance on the validation set was selected for final evaluation on the test set. All experiments were conducted on a workstation equipped with NVIDIA A6000-48GB GPUs.

For the OQMD dataset, we follow the Crystalframer (Ito et al., 2025) to use a larger batch size of 1024 and fewer epochs of 200 on multiple GPU devices.

## C  GENERALIZATION ANALYSIS

To evaluate the generalization ability of our proposed MoCE (Mixture of Crystal Experts) framework across crystal structures of varying complexity, we compared its performance against a baseline model consisting solely of an SE(3)-invariant branch. We grouped the materials in the MP dataset by the number of atoms in their unit cell (Size Group) and statistically analyzed the prediction errors for band gap and formation energy for each group. The results are presented in Fig. 3.

As illustrated in the left panels of Fig. 3, the singular SE(3)-invariant model exhibits significant instability when processing crystal structures of certain sizes. For both the band gap and formation energy prediction tasks, its error shows prominent spikes in several size ranges (e.g., for unit cells with 120-130 or 180-190 atoms). This indicates that a single, fixed geometric inductive bias, such as SE(3) invariance, while effective in many scenarios, struggles to universally handle all types of crystal structures. Its predictive performance deteriorates sharply when encountering structures that do not perfectly align with its underlying assumptions, revealing limited generalization capability.

In contrast, the right panels of Fig. 3 clearly demonstrate the superiority of our proposed MoCE model. Across all crystal sizes, the MoCE model consistently maintains a low mean prediction error, and its error distribution (standard deviation) is substantially smaller than that of the baseline. Crucially, the multiple error spikes observed in the baseline model are effectively suppressed or even eliminated. This result provides strong evidence for the robustness and superior generalization ability of the MoCE framework. By dynamically combining multiple expert networks (e.g., an SE(3) expert for global periodicity and another focusing on local chemical environments), our model can adaptively select the most suitable representation for a given crystal structure, thereby avoiding the limitations of a single-paradigm model when faced with diverse materials. Consequently, this mixture-of-experts strategy successfully enhances the model's performance consistency and predictive accuracy across the entire materials space, validating the novelty and effectiveness of our multi-representation approach.

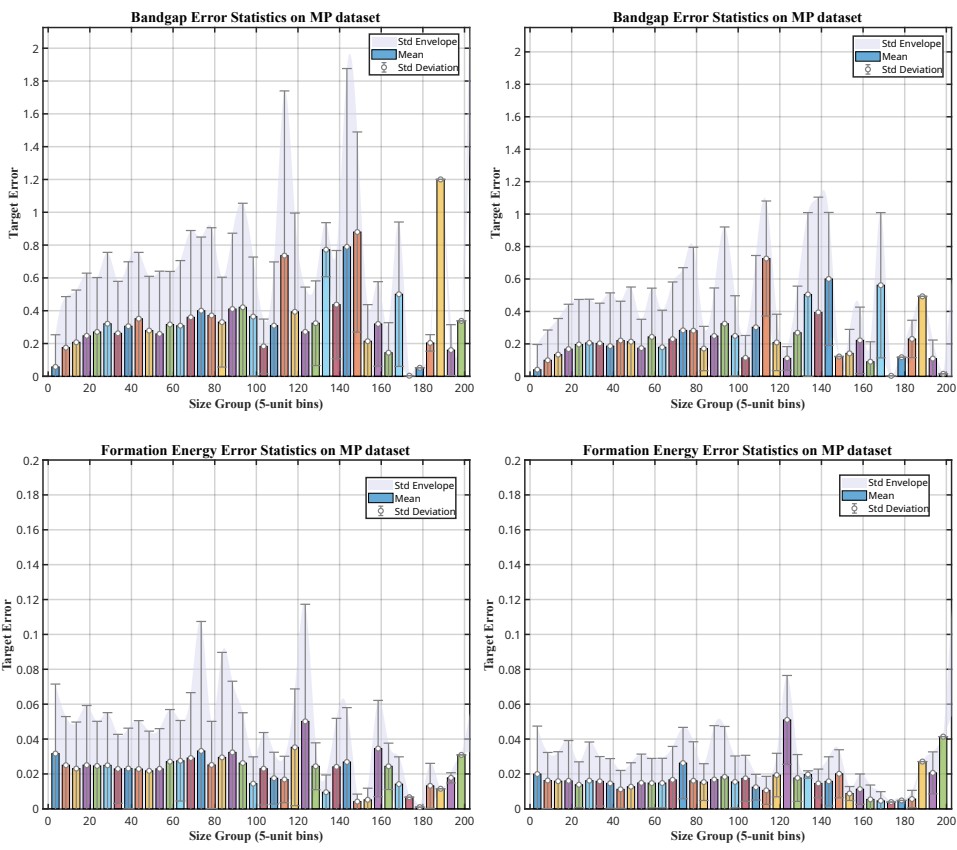

Figure 3: Statistical results of prediction errors in formation energy and band gap for different invariance models under identical architectures and settings on the MP dataset (only SE(3)-invariant on the left, the proposed MoCE on the right).

## D LLM USAGE STATEMENT

According to the ICLR 2026 policy, we hereby disclose our usage of large models in the section: We only used large language models to polish the writing.

