# OpenReview forum: "MoCE: Mixture of Experts Representation for Robust Crystal Material Property Prediction"
_ICLR.cc/2026/Conference — Submitted to ICLR 2026_

### Official Review · Reviewer_pfu4 · 2025-10-26

**Soundness:** 2
**Presentation:** 2
**Contribution:** 1
**Rating:** 2
**Confidence:** 4

**Summary:**

The paper integrates several representations with message passing to enhance prediction performance. Nonetheless, its contribution is weakened by limited originality, unclear exposition, and insufficient acknowledgment of prior work from which key design elements appear to be adopted. Specifically, the SE(3) representation is very similar to iComFormer graph representation that uses bond lengths and three relative angles with respect to periodic repeating patterns, and SO(3) representation is very similar to traditional CGCNN graphs. The authors claim that these different representations capture different perspective of the geometric information, but no theoretical or experimental evidence has been provided.

**Strengths:**

## Strengths

- The overall idea of combining several different representations is interesting.

- The performance of the proposed framework is good, as evaluated on MP, JARVIS and OQMD.

**Weaknesses:**

## Weaknesses

- The proposed SO(3) representation is, in fact, SE(3), despite the authors’ claim. SO(3) invariant representation remains sensitive to translations, which contradicts the stated invariance property.

- The proposed SE(3) representation appears to be identical to that of iComFormer, while the claimed SO(3) representation largely replicates the conventional CGCNN graph formulation.

- The writing lacks clarity in several sections, and certain concepts are introduced without sufficient rigor. For instance, Definition 2 is directly borrowed from ComFormer, yet no proof is provided to demonstrate that the proposed representation is geometrically complete.

- The paper omits comprehensive comparisons with prior methods on Matbench, which limits the strength of its empirical validation.

- The model achieves improved performance at the expense of computational efficiency.

**Questions:**

As listed above in weaknesses.

---

### Official Review · Reviewer_8X2V · 2025-10-27

**Soundness:** 3
**Presentation:** 3
**Contribution:** 2
**Rating:** 4
**Confidence:** 4

**Summary:**

This paper addresses an important task in material design, crystal property prediction, and proposes a mixture-of-experts representation that combines an SE(3)-invariant global structural representation, an SO(3)-invariant local atomic representation, and a dynamic graph module to enhance the accuracy of predicting various benchmark crystal properties.

**Strengths:**

- The paper is very well written, and the authors have effectively motivated the problem. They have provided sufficient background to understand the context and have clearly explained the key challenges in detail, especially in Figure 1.
- The main empirical results demonstrate performance improvements of the proposed methodology on benchmark tasks across three popular datasets.

**Weaknesses:**

- The paper lacks methodological novelty. The modules such as SE(3)-invariant, SO(3)-invariant, and dynamic graph modules are already well established in the literature. The authors primarily combine these existing components within a Mixture-of-Experts (MoE) framework.

- The aspect of multi-modality involving SE(3)-invariant graphs and textual representations has already been explored in CrysMMNet[1], which the authors did not include for comparison in their results.

- In the key challenges section, the authors mention that many material properties of interest—such as elasticity and thermal conductivity—are tensorial and depend on crystal orientation. However, no results are presented to demonstrate that the proposed MoCE performs better on these properties.

- Additionally, MoCE appears inefficient in terms of latency and the number of parameters, raising the question of whether the performance gains are substantial enough to justify the added complexity.

- In the ablation study, it would be interesting to see how the “Only Dynamic Graph” configuration performs on these properties.

- Finally, some of the popular and relevant works[1][2][3] in crystal property prediction are missing from the related work section. In particular, the following papers should have been discussed or cited:

[1] "CrysMMNet: Multimodal Representation for Crystal Property Prediction." Uncertainty in Artificial Intelligence (UAI), PMLR, 2023.

 [2] "CrysXPP: An Explainable Property Predictor for Crystalline Materials." npj Computational Materials, 8(1), 2022, 43.

 [3] "CrysGNN: Distilling Pre-Trained Knowledge to Enhance Property Prediction for Crystalline Materials." Proceedings of the AAAI Conference on Artificial Intelligence, Vol. 37, No. 6, 2023.

**Questions:**

Check the Weaknesses

---

### Official Review · Reviewer_yrEY · 2025-10-30

**Soundness:** 2
**Presentation:** 3
**Contribution:** 2
**Rating:** 4
**Confidence:** 3

**Summary:**

This paper studies the material property prediction problem and proposes a Mixture of Crystal Expert (MoCE) that dynamically integrates multiple, complementary geometric representations. This multi-representation approach achieves a more flexible and powerful framework for a large range of materials property prediction.

**Strengths:**

1. The presentation is easy to follow and clear.
2. This MOCE combines the representation advantages from Comformer, CrystalFormer, etc.

**Weaknesses:**

1. Though 3 experts are well-engineered, they seem to be directly derived from Matformer, CrystalFormer, etc. Therefore, they can have SE(3)/SO(3)/dynamic representations of materials. I have to admit that I do not see much novelty or special designs.
2. The MoE design is naive, and I cannot even consider it as a common MoE architecture.
(a) Firstly, the current design has 3 experts and 1 gating network. The input for the gating network is the outputs of 3 experts. This is different from the MoE design, where both experts and the gating network take the same input.
(b) I understand that the weighted combination provides flexibility and normally better performance. But again, I do not call this MoE architecture.
(c) Have authors explored the sparse MoE architecture? For example, add sparse MoEs before the concatenation of 3 representations? Thus, the architecture will be like input --> 3 representations --> 3 sparse MoEs for different representations --> weighted sum --> head. This will make the name MOCE more suitable.
(d) This weighted sum design, after 3 representations, is also simple. For all these 3 representations, they should have message passing processes. Why not merge the MoE inside the message passing?
3. If authors set the weights to be 1/3, what will be the performance?

**Questions:**

Please check the weakness part

---

### Official Review · Reviewer_wYft · 2025-11-02

**Soundness:** 3
**Presentation:** 3
**Contribution:** 2
**Rating:** 4
**Confidence:** 4

**Summary:**

The paper introduces MoCE (Mixture of Crystal Experts), a novel framework for robust crystal material property prediction that integrates multiple geometric representations through a mixture-of-experts (MoE) approach. MoCE combines three specialized modules: an SE(3)-invariant expert to model global periodic structures, an SO(3)-invariant expert to capture local atomic and molecular conformations, and a dynamic graph network that learns flexible topological relationships beyond static cutoff-based graphs. A gating network dynamically fuses these complementary experts, allowing the model to adapt its representation to the unique structural and chemical characteristics of each crystal.

This unified design resolves the trade-off between global lattice modeling and local chemistry representation that limits conventional crystal graph networks. Experiments on benchmark datasets, Materials Project, JARVIS, and OQMD, demonstrate that MoCE achieves state-of-the-art performance, surpassing prior models such as Crystalformer, ALIGNN, and PotNet in predicting properties like formation energy, bandgap, and elastic moduli. The ablation studies confirm that each expert contributes critically to overall accuracy, while efficiency analysis highlights moderate computational overhead due to its multi-branch architecture. Overall, MoCE provides a powerful and generalizable framework that bridges rigid inorganic crystals and flexible molecular systems, advancing the frontiers of AI-driven materials prediction.

**Strengths:**

The paper is clearly written and well structured, providing a thorough motivation for the problem of crystal property prediction.


The authors present a comprehensive background and effectively explain the key modeling challenges, particularly those illustrated in Figure 1.


The empirical evaluation is extensive, demonstrating strong and consistent performance improvements of the proposed MoCE framework across three widely used datasets — Materials Project, JARVIS, and OQMD.

**Weaknesses:**

The paper offers limited methodological novelty. The main components — SE(3)-invariant, SO(3)-invariant, and dynamic graph modules — are all well-established techniques, and the contribution largely lies in integrating them within a Mixture-of-Experts (MoE) setting.


The work omits comparison with CrysMMNet, which already explores multimodal learning combining SE(3)-invariant graphs and textual representations.



The model exhibits high latency and parameter overhead in Table-5, raising concerns about computational efficiency and whether the observed performance gains adequately justify the added complexity.


- The ablation study lacks a configuration testing the performance of the “Only Dynamic Graph” variant, which could provide deeper insight into that module’s standalone contribution.

**Questions:**

See weakness

---

### Meta-Review · Area_Chair_zdme · 2025-12-13

**Summary:**

Strengths:
* Strong empirical evaluations that demonstrate consistency improvements on material benchmarks: Materials Project, JARVIS, and OQMD


Weaknesses and questions:
* Limited methodological novelty was the reviewer’s shared concern. The paper uses the established components such as SE(3)-invariant, SO(3)-invariant, dynamic graph (reviewer wYft
* Computational efficiency. The model consists of multiple components and introduces the latency and parameter overhead. The reviewers questioned if the additional complexity is necessary (reviewers wYft, 8X2V, pfu4)
* Missing comparisons on CrysMMNet and Matbench (reviewers wYft, 8X2V)
*  MoE design is different from the conventional MoE architecture. Reviewer yrEY asked for additional ablations with simpler choices around MoE.

**Reviewer Concerns:**

The authors did not engage in the discussion

**Reviewer Scores:**

The scores remain unchanged

---

### Decision · Program_Chairs · 2026-01-26

Reject